# Memory of Oneself, Memory of God

Enrique Martínez 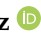

Department of Education and Humanities, Universitat Abat Oliba CEU, E-08022 Barcelona, Spain;
emartinez@uao.es

**Abstract:** Knowledge and love, as well as intellect and will, while remaining distinct, must be understood in man as being intrinsically related on account of his being ordained to a single ultimate end. This responds to the substantial inclination that every human being has for synthesis, for intimate unity. One of the clearest examples of this spirit of synthesis can be recognized in the teaching of Saint Thomas Aquinas; for example, in his attitude of assimilation of the doctrines of many authors, mainly Aristotle and Saint Augustine. We will try to show how, in the thought of Aquinas, faithful to his Augustinian heritage, will and intellect find their rootedness in the memory of oneself, based on the doctrine of the created good according to mode, species, and order. In this the mind is the image of God, to which man is ordered by knowledge and love. Therefore, we can conclude that the memory of oneself is memory of God. We will follow the teachings of the main representatives of the Thomistic School of Barcelona.

**Keywords:** soul; image of God; memory of oneself; intellect; will; memory of God; philosophical theology; Thomism; Augustinianism; Thomistic School of Barcelona

## 1. Introduction

"Love itself is knowledge". This surprising and somehow provocative statement of Saint Gregory the Great (1857, p. 1207) could lead us to confound the operations of the intellect with those of the will, and consequently the nature of the powers themselves. However, nothing could be further from his intention, as is evident from the use that Saint Thomas Aquinas makes of the statement when commenting on the Lord's farewell at the Last Supper: "I have made known to you all that my Father has told me; and so I have called you my friends" (John 15: 15). Aquinas distinguishes between the imperfect knowledge that the apostles possess, and the perfect knowledge that is proper to beatific vision. Faith is therefore a knowledge that moves us to savor the charity that we already possess, and to love the perfect vision that we do not yet possess (Aquinas 1972, c.15, lect.3). Hence, knowledge and love, as well as intellect and will, while remaining distinct, must be understood in man as being intrinsically related on account of his being ordained to a single ultimate end.

This responds to the substantial inclination expressed by the Jesuit Ramón Orlandis in the following terms: "Every man, as man, has an invincible appetite for synthesis, for intimate unity" (Orlandis 2000, p. 369). The Thomistic School of Barcelona, founded by Fr. Orlandis, has always sought in the teachings of the Angelic Doctor a unity according to synthesis as a way of explaining created reality (Canals 2014). One of the manifestations of the spirit of synthesis can be acknowledged in Saint Thomas Aquinas's teaching; for instance, in how he assimilated the doctrines of numerous authors, mainly Aristotle and Saint Augustine.

The avowal of this approach by Canals (2004, p. 51) leads us directly to the specific topic of our paper:

> In all his teachings, our teacher [Ramón Orlandis] insisted on the synthetic unity between the Neoplatonic and Augustinian heritage and the Aristotelian

contribution in the work of the Angelic Doctor. The metaphysics of the spirit contained in the treatise *De Trinitate*, and the doctrine of the good—constituted in creatures as a trace of the Trinity in its *species*, *mode* and *order*, which in spiritual creatures is realized as memory, intellect and will—in which the mind unfolds, knowledge and love (themes not even hinted at in the Twenty-Four Theses) are central to the Christian thought of St. Thomas.

Following this intriguing path mapped out by Canals, we will try to show here how, in the thought of Saint Thomas Aquinas, faithful to his Augustinian heritage (Brock 2006; Cory 2012, 2013; Elders 2015; Wippel 2006), will and intellect find their rootedness in the memory of oneself. Moreover, this memory of oneself is ultimately a memory of God.

## 2. Metaphysical Foundation of Created Good

The text of Canals, which serves as a guide, leads us to the Augustinian doctrine of created good according to mode, species, and order, to account for the distinction between memory, intellect, and will in the mind. It is therefore convenient to begin our argumentation with the metaphysical foundation of all created good, and thereafter, turn our attention to that of the rational creature.

But why start with being as good? Because, as Saint Thomas says: "Good, as a cause, is prior to being, as is the end to the form" (Aquinas 1889, I, q.5, a.2 ad 1). The being, insofar as it is a creature, is a good wanted by the Creator; and without such an intention of the good as an end, there would be no *esse* of the being.

The Augustinian characterization of good as mode, species, and order, to which Canals alludes, helps us to establish the metaphysical foundation for the perfection of the created good, on account of which perfection that good is desirable. Orlandis (2000, p. 377) highlights the importance of this foundation, pointing to its fruitfulness for the explanation of various spheres of reality, such as the one that concerns us today:

> This ternary enumeration, to which the Thomistic synthesis account does not give much importance, in our opinion is one of the main milestones that is absolutely necessary to keep in sight for anyone wanting to follow the holy Doctor in the development of his thought on the ontological, as well as the psychological, moral and spiritual, level.

Following this ternary characterization, three reasons for perfection must be acknowledged in every finite being: mode, which refers to *esse*; species, which refers to form; and order, which refers to the act itself of perfecting (Aquinas 1889, I, q.45, a.7 in c.; O'Callaghan 2007).

Thus, the first and fundamental reason for the perfection of every being is its *esse*. Indeed, *esse* is the actuality of the being, which is perfect because it *is* in act, and it is desirable or good on account of its perfection (Aquinas 1889, I, q.5, a.1 in c.). However, in a created being, *esse* is participated in in accordance with a mode or measure given by its essence, which is related to *esse* as a recipient, is related to that which is received (Aquinas 1889, q.4, a.1 ad 3). A being is therefore desired as a good because of its *esse*. Moreover, the same being radically desires its own *esse*, and hence the common inclination of every being is the conservation of its own *esse* (Aquinas 1892, I-II, q.94, a.2 in c.). Aquinas links this to the conservation of unity (Aquinas 1889, I, q.11, a.1 in c., q.45, a.7 in c.), and if a being is one by its *esse*, it is therefore distinct from any other being by reason of its own *esse*.

The *esse*, insofar as it is participated in according to a measure, inevitably refers to an ultimate final cause in which the *esse* is not found to be participated in; and such is God, the creator, the *Ipsum Esse Subsistens* (Aquinas 1889, I, q.13, a.11 s.c., ad 3; Roszak 2015; Manresa 2017):

> It is written that when Moses asked, "If they should say to me, What is His name? what shall I say to them?" The Lord answered him, "Thus shalt thou say to them, HE WHO IS hath sent me to you". Therefore this name HE WHO IS most properly belongs to God ( . . . ) It is not necessary that all the divine names

should import relation to creatures, but it suffices that they be imposed from some perfections flowing from God to creatures. Among these the first is existence, from which comes this name, HE WHO IS.

The perfection of the divine Being, in its pure actuality, shows that there is no potentiality in Him that would move Him to create, seeking something good external to Him: rather, He creates for the love of His Goodness (Aquinas 1889, I, q.32, a.1 ad 3). Thus, every being must ultimately be explained as having been created by God, from whom it receives its *esse*. Likewise, its unity and distinction are also a participation in divine Unity and Transcendence (Aquinas 1950, c.1, lect.2).

*Esse*, the actuality of all things (Aquinas 1889, I, q.4, a.1 ad 3), is also the cause of every other perfection that can be found in a being, such as its species and order. Let us therefore examine the second reason for the perfection of a being: the form whereby it is determined to a species. The union of the compound is given by the form, since form is the act of matter (Aquinas 1889, I, q.3, a.2 in c.). If the inclination of every being on account of its *esse* is to preserve itself, the inclination that corresponds to form is to communicate itself through some likeness (Aquinas 1889, I, q.196, a.4 in c.). This communication of the form is evident in generation, but the most perfect mode of communication occurs in knowledge. Indeed, in a corporeal substance, what is most significant about the perfection of its form, is that it is knowable in potency; and Aquinas adds that it is known in act by the cognitive subject when it attains the likeness of the form of the being, which is then the form of the knowing intellect (Aquinas 1889, I, q.16, a.2 in c.).

One ought to think about the formation of this likeness not exclusively as an imprinted species, but above all as an expressed species, as the *verbum mentis* that manifests that which is known. The communicativeness proper to the form of a cognitive being consists essentially in manifesting it, as John of Saint Thomas so aptly expressed, faithfully following his teacher: "It is appropriate, then, for understanding, speaking, and producing a mental verb not only out of need, but in order to manifest and communicate" (John of Saint Thomas 1883, d.32, a.4). Indeed, Aquinas (1972, c.1, lect.1; Torrijos 2019) clearly stated this by commenting on the Gospel passage "In the beginning was the Word" (John 1: 1):

> It is clear then that it is necessary to have a word in any intellectual nature, for it is of the very nature of understanding that the intellect in understanding should form something. Now what is formed is called a word, and so it follows that in every being which understands there must be a word.

The aforesaid unity that is found in every being, whereby it is distinguished from any other being, is therefore confronted, in the whole of the Universe, with a multiplicity, which is a sign of imperfection. That the intellect should become one with what it knows is precisely what allows one to overcome this multiplicity, and to reach a cognitive union with all beings: "In this way it is possible for the perfection of the entire universe to exist in one thing" (Aquinas 1976b, q.2, a.2 in c.). However, we ought not to forget that the perfection of this cognitive union is ultimately founded in the *esse* of the being, and not in its form: "The being of the thing, not its truth, is the cause of truth in the intellect" (Aquinas 1889, I, q.16, a.1 ad 3).

Just as the *esse* of the being refers to the first creating Cause, so does its form. This is explained through the likeness that is found in every being with respect to the Divine Essence, although not univocally but analogously (Aquinas 1889, I, q.4, a.3 in c.). Yet, it is not that the Divine Intellect forms a likeness of a created being: rather, a being is created from the form of the Divine Intellect, which is nothing other than the Divine Essence (Aquinas 1889, I, q.15, a.2 in c.).

Let us now turn to the third reason for perfection of the created being, which is the ordination to an end and to what is ordered to the end. Thus, the inclination to an end follows the form: "Upon the form follows an inclination to the end ( . . . ); and this belongs to weight and order" (Aquinas 1889, I, q.5, a.5 in c.). This ordering to an end according to a proper form must be understood as an ordering to a proper perfection (Aquinas 1892, I–II,

q.1, a.6 in c.). This inclination to a proper perfection, to state it again, has its cause in the *esse*, although the form is now presupposed.

Nonetheless, this perfection, to which one tends, is still in potency; and *good* conveys actuality and perfection. Therefore, every being requires, for its own perfection, other beings that, as goods in act, help it to attain its deepest natural inclination. For this reason, St. Thomas explains that beings seek a union with each other, which is not a cognitive one in the intellect, but a union with the thing itself, and this alleviates—in a certain fashion, although in another way—the imperfection of the multiplicity of beings of the Universe: "Those things that are absolutely different in the creatures are somewhat united by a certain order, so that, at least, they imitate the Unity of God" (Aquinas 1950, c.1, lect.2).

This last reference to the imitation of divine Unity leads us back to the ultimate cause of created being—in this case, under the *ratio* of end. Indeed, the order of every being to its proper perfection, always finite and desired as potential, demands an ultimate and desirable end in its perfect actuality. However, this ultimate end is nothing other than the first cause of the *esse*, who created the beings to liberally communicate His likeness. Thus, all beings tend to attain their perfection, which consists in resembling divine Goodness; whence, "the divine Goodness is the end of all things" (Aquinas 1889, I, q.44, a.4 in c.).

If every being exists virtually in God as the cause of its *esse*, which He creates, and in His Intellect as the cause of its truth, which He knows, we must conclude that it exists in Divine Will as the cause of its good, which He loves (Aquinas 1889, I, q.20, a.2 in c.). However, God does not love created beings as one who is moved by its good: rather, for the sake of His perfect Goodness, "the love of God infuses and creates goodness in things" (Aquinas 1889, I, q.20, a.2 in c.).

## 3. Memory of Oneself Becomes Memory of God

We now proceed to set forth the metaphysical foundation of the mind from the triad of mode, species, and order, bearing in mind what Canals says in the text we are following: "The good [is] constituted in creatures as a trace of the Trinity in its species, mode and order, which in spiritual creatures is realized as memory, intelligence and will" (Canals 2004, p. 51).

As already noted, *esse* is the actuality of a being, whereby it is perfect and good. *Esse* is therefore the actuality also of the intellectual creature and of all its perfections. Yet, the mode or measure in which *esse* is received, is in this case very different from that of other beings. Indeed, the intellectual creature participates in *esse* without dependence on its matter. This participation is known as "to subsist" or, as Canals calls it, "the subsistence of the act in itself" (Canals 1987, p. 483). Such independence occurs even when it entails, in man, the substantial union of the soul with the body—although "This *esse*, insofar as it is of the soul, is not dependent on the body" (Aquinas 1976a, c.3).

Not depending on matter allows the intellectual creature, both angelic and human, to participate in *esse* according to the *ratio* of infinity (Aquinas 1889, I, q.7, a.1 in c.). Again, St. Thomas refers this to the human intellect: "The intellect is still further cognitive, because it is more separated from matter and unmixed" (Aquinas 1889, I, q.14, a.1 in c.; Canals 1987, pp. 498, 601–2).

We turn now to what this subsistence—or proper mode of the intellectual creature—implies in its independence from matter. Saint Thomas, citing the *Liber de causis*, characterizes it as a "return to itself" (Aquinas 1889, I, q.14, a.2 ad 1):

> Returning to its essence means nothing other than something that subsists in itself. For the form, insofar as it perfects matter by giving it its *esse*, somehow spills over it; but insofar as it has *esse* in itself, it returns to itself.

According to Canals, Bofill constantly repeated this text, which is indeed the core of the Augustinian-Thomistic metaphysics of the spirit, which we are outlining here, following the teachings of the Thomistic School of Barcelona.

We must, then, further explore the meaning of this *reditio in seipsum*. If in every being there is an inclination to preserve itself in existence, the act of subsisting can only be understood in terms of the intellectual operations—knowledge and love—in such a way

that these very operations will, in turn, be illuminated by the act of subsisting. Aquinas resolves this in the afore-quoted text from a cognitive perspective: "Cognitive faculties that are not subsistent but are acts of some organs, do not know themselves, as is evident in each of the senses. But the cognitive faculties subsisting by themselves know themselves" (Aquinas 1889, I, q.14, a.2 ad 1).

*Returning to oneself* means, therefore, *knowing oneself*. However, we must bear in mind that the soul can know itself in two ways, as Aquinas teaches, again following Saint Augustine: in one way, the soul understands itself in its essence through a concept that can be universally predicated of every soul; in another way, one knows oneself "insofar as it has *esse* in such individual" (Aquinas 1976b, q.10, a.8 in c.), that is, one's own singularity, perceiving that one has *esse*.

Thus, this knowledge of oneself in respect of one's own *esse* is what corresponds to the act of subsisting according to the mode of the intellectual creature. This knowledge is attained, therefore, not according to the mode of an object, but according to the mode of the presence of one's own *esse*: "The mere presence of the soul suffices ( . . . ) which is the principle of the act by which it knows itself. That is why it is said to know itself by its own presence" (Aquinas 1889, I, q.87, a.1 in c.). Bofill calls this mode of knowledge according to presence "memory": "We will call *memory* the function of presentiality that is implied in this same cognitive relation, considered in order to its principle" (Bofill 1967a, p. 99; Canals 1987).

We finally come to the term that forms the backbone of our argument, which is "memory", in the sense used by Saint Augustine. Indeed, after asking: "In what way would it not be present to itself when it is not thinking of itself, for it can never be without it?", he answers: "When it is not thinking of itself, it would indeed not be in its own presence, nor would its own perception be formed from it, and yet it would know itself as though it were to itself a memory of itself" (Augustine of Hippo 1865, p. 1042). This *memoria sui* should not be understood as a faculty comparable to the intellect and the will. On the contrary, *memory* refers directly to the very *esse* of the intellectual creature that, unencumbered by matter, is evident to itself in the subsistence of the act in itself. As Bofill (1967b, p. 71) explains: "Memory is here not so much a particular faculty of the subject, but the general faculty of self-presence". Since one's own *esse* is habitually present to oneself, it can be identified as "memory", that is, memory of oneself: memory of one's own *esse*. Memories of past events will only be possible on account of a radical recognition of one's own identity subsisting in the memory of oneself (Canals 2004).

Again, this is a habitual knowledge of one's own being, for existential knowledge of oneself is twofold: habitual and actual. The latter occurs in the exercise of cognitive or appetitive operations (Aquinas 1976b, q.10, a.8 in c.). Habitual knowledge, on the contrary, is attained by the very presence of the soul to itself; and just as memory is not a faculty, as already noted, the term "habitual" does not mean here a habit characterized by its potentiality, but the very essence of the soul characterized by its actuality. The soul is always present to itself, even when it is not knowing itself in act (Aquinas 1976b, q.10, a.8 in c.). For the same reason, memory of oneself is essentially "memory in act" (Canals 1987, pp. 470–74): "The mind itself is intelligible in act" (Aquinas 1976b, q.19, a.6 in c.).

The *esse* of every being supposes, as already noted, unity and distinction. The *esse* of the intellectual creature even more so. It is precisely the memory of oneself that reveals the unity and distinction of a being that subsists in an intellectual nature; wherefrom, such a being has been given a name that signifies this singularity: that of "person" (Aquinas 1889, I, q.27, a.3 ad 2). Therefore, the memory of oneself must be understood as an intimate and personal knowledge, rather than a subjective one. Whence, Canals (2004, pp. 77–78) states:

> If we were not to recognize this existential consciousness that belongs to each man on account of his *esse*, nor would we be able to explain daily human sociability, or family or popular traditions; or to explain that, autobiographical memories such as the *Confessions* of Saint Augustine, should have been written (to quote only a prime example); nor would knowledge of history have a contemplative signifi-

cance; nor could, in literature, so many enriching writings of human knowledge have emerged: novels and plays, attending primarily and profoundly to human life in what is singular and existential in it.

The *esse* of the intellectual creature, according to its mode of participation, refers—as does every being—back to God the Creator, *Ipsum Esse subsistens* (Contat 2011). In this case, however, God is the ultimate cause of a subsistent being that has a memory of itself. This means that the purpose of the Creator regarding intellectual subsistents is precisely to make them participants—in some measure—of the knowledge that He has of himself. As St Thomas says, "It supremely belongs to God to be self-subsisting ( . . . ) He supremely returns to His own essence, and knows Himself" (Aquinas 1889, I, q.14, a.2 ad 1). Every intellectual creature, therefore, knows itself in a way that is similar to that by which God knows himself. Saint Thomas then adds that this allows the intellectual creature not only to turn back to itself, but even to return to God, because "an effect reaches all its perfection when it returns to its beginning" (Aquinas 1961, II, c.46, n.2). Memory of oneself becomes, then, memory of God.

Such is, therefore, the condition of each intellectual creature in its singularity, that it must be said that God has created it for the sake of itself; and hence, it is a person, as God is personal, "in a more excellent way" (Aquinas 1889, I, q.29, a.3 in c.). Wherefrom, St. Thomas states that intellectual creatures are the constitutive part of the Universe, while irrational ones are only accidental parts of it (Aquinas 1961, III, c.112, n4).

Consequently, we conclude this section by stating that the intellectual creature—in its mode of participating in *esse* as a subsistent being that has memory of itself—is the maximum good amongst all created goods of the Universe because it has been loved by God—in the creating act—as a good in itself, to which His own goodness can be communicated. Precisely for this reason, it is also maximally true insofar as it can know itself, as Canals says (Canals 1987, p. 576):

> We must uphold the primacy of the spiritual subsistent, of the person, not only in the line of transcendental good (in which personal being expresses the good in an heterogeneous way, diverse from that of any non-personal being, for only a person is capable of being the terminus of love of friendship, capable of corresponding in friendship, and of communicating in the works of life), but also with regard to transcendental truth.

### 4. Memory, Intellect, and Will: Image of God

We have identified in the memory of oneself the mode in which the intellectual creature participates in *esse,* and how the memory of oneself becomes memory of God. However, we must still relate this memory to the intellect and the will, as we originally set out to do, on the basis of mode, species, and order, the triad that corresponds to every created good. To accomplish this, we will start from the two modes of assigning the aforesaid triad to the mind, following the distinction that Saint Thomas draws from Saint Augustine: first, as *mens, notitia, et amor*; then, as *memoria, intelligentia, et voluntas*. Both theologians consider the former to be an imperfect image of the Trinity of God insofar as they convey a relation to habit and to potency, while they consider the latter to be a perfect image insofar as they convey a relation to act (Aquinas 1976b, q.19, a.3 in c.).

Let us first consider the imperfect image. What is imperfect in the triad *mens, notitia, et amor* is its signifying potentially or virtually, as in their principles, the cognitive and volitional acts proper to the perfect image (Aquinas 1889, I, q.93, a.7 in c.). In this way, *mens* generically means the powers of the soul, while *notitia* and *amor* refer specifically to the intellect and the will as principles of their respective acts (Aquinas 1976b, q.10, a.3 in c.).

On the other hand, if *memoria, intelligentia, et voluntas* should convey actuality, then they must be understood as acts and not as powers. Therefore, "memory" designates here the subsistence of the mind insofar as it is present in act to itself, as already explained. At the same time, however, "memory" refers in common to *mens, notitia, et amor* insofar as the aforesaid presence is habitual. Therefrom, saint Thomas somewhere calls *mens, notitia, and*

*amor, "habitus consubstantiales"* (Aquinas 1929, I, d. 3, q. 5, a. 1 ad 2). In this way, *mind*, as well as *knowledge* and *love*, signify the essence itself of the subsistent spirit insofar as it is habitually present to itself; they are not in this case three distinct realities but three different expressions of the same reality.

*Mens* expresses the spirit, referring directly to its subsistent *esse* according to its own mode of participation. Indeed, Aquinas says that although "mind" signifies the intellectual power, it can also designate the essence itself of the subsistent spirit: "Or, if it names the essence, this is only insofar as such a power flows from it" (Aquinas 1976b, q.10, a.1 in c.).

*Notitia*, on the other hand, refers to the mind itself insofar as it is intelligible to itself according to its own species. Indeed, the mind is intelligible in act, as already stated; wherefrom, it possesses its own *notitia* or species of itself without the need for a distinct intelligible species in order to know itself according to its *esse*. As Aquinas says, because of this, "the mind, prior to the reception of images, has habitual knowledge of itself by which it can be perceived as existing" (Aquinas 1976b, q.10, a.8 ad 1). Shortly afterwards, he confirms the distinction between the act of knowing oneself, which is an accident, and knowledge identified with the substance itself of the mind: "Whence, Augustine, in *De Trinitate* IX, says that the *notitia* substantially inheres in the mind insofar as the mind knows itself" (Aquinas 1976b, q.10, a.8 ad 14). Therefore, we say that the intellectual subsistent is true—more indeed, that it is maximally true—insofar as it is intelligible in act to itself, and is capable, due to such intelligibility, to thereafter know itself in act.

*Love*, finally, refers to the mind itself insofar as it is lovable to itself according to its own order or inclination. Whence, we say that the intellectual subsistent is good—more indeed, that it is maximally good—insofar as it is lovable in act to itself and is therefore capable of thereafter loving itself in act.

In short, the triad *mens, notitia, et amor* signifies the memory of oneself, "for we are, we, we know that we are, and we love this being and knowledge", as Saint Augustine says (Augustine of Hippo 1864, p. 339). Canals (2004, p. 110) explains this triad of memory of oneself as mind, knowledge, and love, as follows:

> What mode, species and order is in every being, is the mind—spiritual substantiality——, knowledge—the constitutive and radical capacity for self-consciousness that makes it also capable of knowing the other—, and love—understood here not as a concrete act of affective union or of choice of certain wanted objects, but as the complacency in the very prior and foundational relation of any attainment of a concrete good.

Here, the knowledge and love of memory not only convey a relation to knowledge and love of oneself, but also to knowledge and love of other things. For *mind* signifies the essence itself of the mind: "But if it names the essence, that is only insofar such power flows from it" (Aquinas 1976b, q.10, a.1 in c.), as already noted. That is to say, the intellectual powers and their respective acts emanate from the memory of oneself precisely insofar as knowledge and love of oneself are found in it. Referring to this primacy of the memory of oneself with respect to the intellectual powers, Canals (1987, p.476) clearly states the thesis that we herein defend:

> Guided by the thought of saint Thomas Aquinas, we have come to find the originating unity of all human cognitive faculties in *subsistence in itself* or *immaterial possession of esse*; due to its ontological structure of sameness constitutive of its habitual self-consciousness, and thereby destined to an actual, illuminating and open knowledge of itself in respect of universal being, it is in the mind or intellective soul, *insofar as it possesses esse in itself*, where we find the originating principle and the actuality or perfection to which all cognitive faculties and their proper acts are ordered.

Why does memory have this primacy? Because the *ratio* of *esse* as act, the subsistence of the act of oneself, belongs to it, since "*esse* is itself the actuality of all things and even of forms themselves" (Aquinas 1889, I, q.4, a.1 ad 3). Therefore, any actuality of the intellect

in act, or of the will in act, which in creatures belong to an accidental genus, are rooted as in their principle upon the actuality of the intellectual subsistent, whose *esse* is present to itself according to the mode of memory of oneself. Therefore, the same must be said of the powers that are ordered to their proper acts, that is, the intellect and the will, which, again, belong in creatures to an accidental genus, and which are presupposed in substantial knowledge of oneself and in substantial love of oneself.

Let us pause for a moment to examine the operations of the intellect and of the will in order to better understand their rootedness in memory of oneself. As already discussed, every being tends to communicate the form that it possesses. The most perfect mode of such communication is found when the knower, having formed a likeness of the thing in the intellect, becomes one with the thing known and manifests it in a mental word. For this to occur, it is certainly necessary, in man, that the intelligible in potency be made intelligible in act by abstracting the species from material conditions (Aquinas 1889, I, q.79, a.4 ad 4). Not only that, it is even more necessary, as a prior and foundational condition, that the intellect be intelligible in act, so that the mental word—which makes the *esse* manifest—may arise, as Saint Thomas says (Aquinas 1961, IV, c.14): "Like the origin of act from act, as is brilliance from light and an understanding understood from an understanding in act".

The active intellect itself must therefore be understood as the very spiritual substance, which—in its intelligibility in act—illuminates the images to form the intelligibles in act (Aquinas 1976b, q.10, a.6 in c.). On the other hand, for the will to be able to want a good in act and to seek a certain affective union with it—whether with love of concupiscence or in the most perfect union of friendship—the prior and founding appetibility in act of one's own *esse* is required. This *amor sui*, present to the subsistent spirit, is like a *pondus* or substantial weight that inclines the will and its whole act of willing, so that the will, even though following upon the intellect in its operation, must nevertheless be said to arise from the very substance of the soul: "The will does not come directly from the intellect, but from the essence of the soul, presupposing the intellect" (Aquinas 1976b, q.22, a.11 ad 6). This *pondus* moves the practical and elective judgment itself of the intellect from its origin, for, as Aristotle had already expressed: "The intellect itself, however, moves nothing, but only the intellect which aims at an end and is practical" (Aristotle 1925, VI, c. 2, 1139a 35).

Finally, let us not forget that the intellectual creature is capable of returning to itself due to its act of subsistence, which is not limited by matter, as discussed from the outset. We must therefore conclude, in synthesis, that the intellectual subsistent experiences all its cognitive and appetitive operations as its own, present at all times to its substantial memory. And in this way, it is possible to answer the initial question raised by Berdiaef: Who knows and loves? He who has memory of himself.

We conclude this part with the following words by Martín Echavarría (2013, p. 308), which uphold the thesis taught by the Thomistic School of Barcelona presented here:

> Thus, we could perhaps say that substantial knowledge and love (that is, the spiritual substance as intelligible and lovable in act) are precisely the condition of possibility and the root from which these powers emanate (which are indeed accidents in a creature). These powers, due to their reflective nature, always have their substantial origin present in their operations. Therefore, in every act of the intellect, besides the object, the mind perceives or feels itself as the root of the act, and in every act of love the will does not cease to have itself present as something radically loved.

## 5. Memory of God as *Sapientia Cordis*

We have been able to ascertain, from the characterization of created good as mode, species, and order, the rootedness of the intellect and the will in memory of oneself. However, as indicated from the beginning, the most complete, unitary understanding of all these parts should be attained by considering their ordering in relation to the ultimate end.

This end (by which the Universe is created, and within it the most perfect being, which is the personal subsistent) is divine Goodness Itself. Whence, all cognitive and

appetitive operations, rooted in the memory of oneself, are ordered to attain, in glory, the contemplation of God. There, knowledge and love are united in synthesis, as per Saint Gregory, whose words we quoted at the outset: "Love itself is knowledge". In this contemplation, *memoria sui* will have returned to its ultimate origin as *memoria Dei*. Then, we will recognize ourselves in the knowledge and love that God has for us, as Saint Juan de la Cruz (2006, B, c.38, n.3) teaches, commenting on the Apostle:

> The soul will know God as it is known of Him: *Then I shall know even as I am known*. That is, *I shall then love God even as I am loved by Him*. For as the understanding of the soul will then be the understanding of God, and its will the will of God, so its love will also be His love.

The whole perfective dynamism of human life is therefore characterized by the disposition of oneself to such contemplation, anticipating it as much as possible. This is attained above all in true friendship, in a union that is not only affective, but of intimate, mutual confidence concerning one's personal life. Bofill (1950, pp. 164–65) writes enthusiastically:

> By it [sc., by friendship] is loneliness finally overcome, equally satisfying both our aspiration to be understood, appreciated, loved, on one hand, and, on the other, satisfying those that flow in the opposite direction: to pour out into others the fullness of our heart in quiet confidence. Through them both, man is placed in his true environment, namely, the family and society, and occupies his place in the Universe. What will give us a glimpse of the measure of this perfection and its corresponding joy will be the consideration of what it means: the enrichment of a Person by what is most valuable in the entire Universe—namely, by another Person, who gives himself not in any of his aspects or more or less external goods, but introducing us into the intimacy of his own life and *esse*.

The act itself of imparting true philosophical teaching must be understood as a form of philosophical friendship. Therefrom, Canals used to say that "there is nothing less suitable to *science* and *philosophy* than its qualification as *solitary knowledge*" (Canals 1987, p. 681). This is how the history of the Thomistic School of Barcelona is understood, gathered around the teachings not only of St. Thomas, but even more so, of Christ's Heart. It is not surprising, therefore, that, engraved in the memory of Canals, remained the words (which he often repeated it to his friends) attributed to Saint John XXIII characterizing the doctrine of Saint Thomas: *Sapientia Cordis*. As Canals (2004, p. 85) explains:

> Saint Thomas insistently upholds [the existence of . . . ] a knowledge by connaturality with the thing known [ . . . ]. If this connaturality, rooted in existential consciousness, together with what is apprehended by man, should not exist, we would not know the goodness of beings—neither in practice nor theoretically. John XXIII described the doctrine of the Angelic Doctor as *wisdom of the heart*. A reading of saint Thomas, carried out with a sense of connaturality with him, shows that this knowledge does not invalidate conceptual judgments—rather, it vivifies them and fills them with the plenitude of their meaning.

*Sapientia Cordis*, the Wisdom that flows from the Heart, synthesizes the rootedness of intellect and will in memory of oneself as memory of God.

**Funding:** This research received no external funding.

**Data Availability Statement:** Not applicable.

**Conflicts of Interest:** The author declares no conflict of interest.

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
