# Peer review of "Memory of Oneself, Memory of God"

_religions, doi:10.3390/rel13060567_

Round 1

Reviewer 1 Report

Dear author, as a person who knows little about Thomas Aquinas' cognitive philosophy and as a person who loves scientific models which help us visualize the structures of the created world, I missed some symbolic or analog model I would have in front of my eyes all the time and which would help me make sense of the text I read. Many conceptual connections have been made in the text, the meaning and validity of which I find difficult to judge because they are not related to any trusted model or lifeworld experience known to me. Maybe these connections are appropriate, maybe not, in the end I was left unsure and without a real idea of ​​what I had actually learned anew. I miss some more concrete connection to lifeworld experience that would make all these abstract connections more perceptible or graspable. This is, of course, my subjective view from my perspective, which is certainly too demanding to be able to suggest any concrete changes to your article. I allow the possibility that your text is meaningful and justified to other experts, who know more about the topic.

Author Response

Dear reviewer

I am very grateful for your sincere comments. Following your suggestion, I will propose to the editors of the journal to send the article to another reviewer so that he/she can evaluate it. 

Best regards

Reviewer 2 Report

See the attached PDF with comments on the body of the document for some minor edits that are needed

Author Response

I am very grateful for your comments. The suggested corrections are very appropriate, and I will introduce them into the text. 

Best regards

Reviewer 3 Report

This paper aims to argue an original thesis: the connection between the memory of oneself and the memory of God, in the intellectual heritage of  Thomas Aquinas and Augustine.

The metaphysical knowledge of the author is well proven.

However, the methodologies carried out in order to support the main thesis appear as not adequate:

  • the only contemporary references are from Spanish environment. There are a lot of studies about memory in Augustine, the doctrine of esse in Thomas etc by non-Spanish authors. But they are not even mentioned in the paper.
  • The theme of memory, which is preeminent in Augustine, remain absolutely secondary in Aquinas' thought. But this issue is not addressed in the paper.
  • The theoretical structure is not clear in some passages. For example, the use of Aquinas's doctrine of esse to prove something which primarily entails the (augustinian) theme of memory. Or the paragraph about sapient cordis: which is its role in the general theoretical framework of the paper? By the way, the description by John XXIII of Aquinas's thought as "sapientia cordis" is not well proved. There are no references of that, apart from the declaration by Canals himself and a short reference I found here: https://www.jstor.org/stable/45075900. But it is just a short and not direct passage by Pope John XXIII.

Author Response

I sincerely appreciate the comments made on my article. I will try to respond to them, indicating the modifications made.

  • The only contemporary references are from Spanish environment. There are a lot of studies about memory in Augustine, the doctrine of esse in Thomas etc by non-Spanish authors. But they are not even mentioned in the paper. the only contemporary references are from Spanish environment. There are a lot of studies about memory in Augustine, the doctrine of esse in Thomas etc by non-Spanish authors. But they are not even mentioned in the paper.
  • Answer: Indeed, the main references provided are from Spanish environment -except for the studies by A. Contat, J. O'Callaghan and P Roszak-. This is justified in this way: as the reviewer rightly points out, "The theme of memory, which is preeminent in Augustine, remain absolutely secondary in Aquinas' thought”; one of the relevant contributions of the so-called "Thomistic school of Barcelona", mainly with R. Orlandis, J. Bofill and F. Canals, has been the rediscovery of the Augustinianism of Thomas Aquinas, above all with regard to the theme of memory and self-knowledge. This is the reason why the present study has focused on these authors. Nevertheless, the bibliography has been completed with contributions from other authors outside the Spanish environment, who are close to this perspective: Therese Cory, Leo Elders, Stephen L. Brock and John Wippel.

  • The theoretical structure is not clear in some passages. For example, the use of Aquinas's doctrine of esse to prove something which primarily entails the (augustinian) theme of memory.
  • Answer: Section 2 of the article tries to link Aquinas' metaphysics of esse with the Augustinian doctrine of the good as mode, species and order. This is the main novelty when it comes to understanding the Augustinian memoria sui as cognitio de anima secundum quod habet esse in tali individuo of Aquinas. This is what is done in the third section. The synthesis of what has been said can be found in these affirmations: “This knowledge of oneself in respect of one’s own esse is what corresponds to the act of subsisting according to the mode of the intellectual creature. This knowledge is attained, therefore, not according to the mode of an object, but according to the mode of the presence of one’s own esse (…) Bofill calls this mode of knowledge according to presence memory” (212-221).

  • Or the paragraph about sapient cordis: which is its role in the general theoretical framework of the paper?
  • Answer: The role of this paragraph is to show the link between the metaphysics of esse and the notion of the good, linking Augustine with Aquinas. Sapientia Cordis comes to show that wisdom is not adequately understood without love. This is a response to the initial affirmation of the article: “Love itself is knowledge”

  • By the way, the description by John XXIII of Aquinas's thought as "sapientia cordis" is not well proved. There are no references of that, apart from the declaration by Canals himself and a short reference I found here: https://www.jstor.org/stable/45075900. But it is just a short and not direct passage by Pope John XXIII.
  • Answer: I accept the critical comment, as the statement is not sufficiently substantiated. I therefore add that the statement was merely attributed to John XXIII.

Thank you